# The Utility of Bedside Assessment Tools and Associated Factors to Avoid Antibiotic Overuse in an Urban PICU of a Diarrheal Disease Hospital in Bangladesh

**DOI:** 10.3390/antibiotics10101255

**Published:** 2021-10-15

**Authors:** Farzana Afroze, Md. Tanveer Faruk, Mehnaz Kamal, Farhad Kabir, Monira Sarmin, Mithun Chakraborty, Md. Rezaul Hossain, Shamima Sharmin Shikha, Visnu Pritom Chowdhury, Md. Zahidul Islam, Tahmeed Ahmed, Mohammod Jobayer Chisti

**Affiliations:** Nutrition & Clinical Services Division (NCSD), International Centre for Diarrhoeal Disease Research, Bangladesh (icddr,b), Dhaka 1212, Bangladesh; tanveer.faruk@icddrb.org (M.T.F.); mehnaz.kamal@icddrb.org (M.K.); md.farhad.kabir@gmail.com (F.K.); drmonira@icddrb.org (M.S.); drsharif@icddrb.org (S.); mithunnemc@gmail.com (M.C.); hossain.rezaul@icddrb.org (M.R.H.); shamima.sharmin@icddrb.org (S.S.S.); visnu.pritom@icddrb.org (V.P.C.); zahidul.islam@icddrb.org (M.Z.I.); tahmeed@icddrb.org (T.A.)

**Keywords:** antibiotics, PICU, qSOFA, qPELOD-2 score, children, diarrhea

## Abstract

Background: Antibiotic exposure in the pediatric intensive care unit (PICU) is very high, although 50% of all antibiotics may be unnecessary. We aimed to determine the utility of simple bedside screening tools and predicting factors to avoid antibiotic overuse in the ICU among children with diarrhea and critical illness. Methods: We conducted a retrospective, single-center, case-control study that included children aged 2–59 months who were admitted to PICU with diarrhea and critical illness between 2017 and 2020. Results: We compared young children who did not receive antibiotics (cases, n = 164) during ICU stay to those treated with antibiotics (controls, n = 346). For predicting the ‘no antibiotic approach’, the sensitivity of a negative quick Sequential Organ Failure Assessment (qSOFA) was similar to quick Pediatric Logistic Organ Dysfunction-2 (qPELOD-2) and higher than Systemic Inflammatory Response Syndrome (SIRS). A negative qSOFA or qPELOD-2 score calculated during PICU admission is superior to SIRS to avoid antibiotic overuse in under-five children. The logistic regression analysis revealed that cases were more often older and independently associated with hypernatremia. Cases less often had severe underweight, altered mentation, age-specific fast breathing, lower chest wall in-drawing, adventitious sound on lung auscultation, abdominal distension, developmental delay, hyponatremia, hypocalcemia, and microscopic evidence of invasive diarrhea (for all, *p* < 0.05). Conclusion: Antibiotic overuse could be evaded in PICU using simple bedside screening tools and clinical characteristics, particularly in poor resource settings among children with diarrhea.

## 1. Introduction

Appropriate use of antimicrobial agents is the cornerstone of effective antimicrobial stewardship programs and has become a focus of patient safety, quality assurance, and health care outcomes. The Intensive Care Unit (ICU) deals with critically ill patients, and infections are among the most frequent causes of hospitalization among patients in the ICU. Thus, antibiotic burden in ICU is very high; 66 to 77% of all ICU patients and 84–100% with an ICU stay of more than 48 h are exposed to at least one class of broad-spectrum antibiotic [1,2]. As the definitive diagnosis of infection is often quite hard, Pediatric ICU (PICU) physicians are concerned about delayed treatment; consequently early initiation of a broad-spectrum antibiotic is expected, especially in the PICU settings [3,4].

However, 50% of antibiotic prescriptions were inappropriate and could be avoided [5,6]. The situation is even worse in Low and Middle Income Countries (LMICs), where prescribing practices are poor, and antibiotic stewardship programs are frequently non-existent [7]. This inappropriate exposure upsurges the emergence of antimicrobial resistance leading to prolonged hospital stays and increases mortality as well as healthcare costs [8,9].

Improving diagnostic tools, and discriminating against bacterial infections from viral infections and other non-infectious mimics, is the eventual solution to reducing unnecessary antibiotic initiation in the ICU. However, bacterial isolation by culture and sensitivity, the gold standard test, is time-consuming and has low sensitivity for slow-growing and fastidious microorganisms [10]. Although molecular diagnostic assays allow faster viral recognition, more rapid bacterial identification, and determination of antimicrobial susceptibilities [11,12,13], these are expensive and not always available in LMIC.

Several scoring systems have been developed to predict suspected infection or sepsis as a surveillance tool, and a positive score might be of value to identify infection promptly. Historically, the systemic inflammatory response syndrome (SIRS) criteria were considered to be fundamental to the diagnosis of inflammation and infection [14], although it might not work in children with dehydration, especially with diarrhea [15]. Subsequently the quick Sequential Organ Failure Assessment (qSOFA) score (altered mentation, systematic hypotension and tachypnea) [16,17], and quick Pediatric Logistic Organ Dysfunction Score-2 (qPELOD-2) (altered mentation, systematic hypotension and tachycardia) [18,19] have been recommended as a handy bedside tool to promptly recognize patients with infection who might be at risk of poorer outcomes and could be predominantly useful in assessing critically sick patients. However, the utility of negative scores has not been validated to identify suspected non-bacterial infectious patients at the ICU who would benefit from the watchful waiting approach.

Informative studies regarding whether clinicians could adopt a ‘watchful waiting’ approach safely and confidently for a subset of PICU patients in LMIC are still lacking. ICU physicians seldom consider a watchful waiting approach; in contrast, they prefer to overuse broad-spectrum antibiotics when faced with a non-reassuring characteristic [20]. Thus, objective and evidence-based criteria are crucial to avoid unnecessary antibiotic exposure and, hence, tribulations. Therefore, the study has two main objectives: (1) to evaluate the accuracy of negative SIRS, qSOFA, and qPELOD-2 scores in predicting no antibiotic or watchful waiting approach, and (2) to identify the associated factors of implementing a no antibiotic approach in the PICU.

## 2. Results

### 2.1. Characteristics of Participants

In total, 2089 children aged 2 months to 59 months were admitted to the ICU of Dhaka Hospital between 2017 and 2020; 268 (12.8%) children did not receive any antibiotics, and we excluded 104 children because they were treated with antibiotics at the emergency ward or after being transferred to another ward.

A final cohort of 510 children formed the analyzable data, where 164 were cases (no antibiotic group), and 346 were controls (antibiotic group).

Bivariate analysis revealed that the cases were more often older, presented with acute watery diarrhea, history of vomiting, seizure, and hypernatremia than the controls (Table 1).

In contrast, cases less often presented with SAM, severe underweight, severe stunting, dehydration, age-specific fast breathing, lower chest wall in-drawing, hypoxemia, adventitious sound on lung auscultation, abdominal distension, altered mentation, developmental delay, and congenital heart disease. The mortality rate was significantly lower amongst the cases compared to controls (Table 1).

Similarly, hyponatremia, hypokalemia, hyperkalemia, hypocalcemia, leukocytosis, and microscopic evidence of invasive diarrhea were less frequent among cases than their corresponding peers (Table 2). Laboratory evidence of CNS infection and bacteremia were nil among cases. Similarly, bacterial pathogens isolated from stool culture were less frequent among cases than controls.

### 2.2. Utility of Bedside Assessment Tools to Prevent Antibiotic Overuse

Of the included children, the frequency of Systemic Inflammatory Response Syndrome (SIRS), quick Sequential Organ Failure Assessment (qSOFA) score and quick Pediatric Logistic Organ Dysfunction Score-2 (qPELOD-2) negativity was 311/510 (61%), 452/510 (89%), and 475/510 (91%), respectively. 

Cases more often had SIRS, qSOFA, and qPELOD-2 negative children compared to controls (Figure 1).

For predicting a no antibiotic approach, a negative qSOFA and a negative qPELOD-2 both displayed a very high sensitivity of 95.9% (95% CI 91.8 to 98.3) and 95.9% (95% CI 91.8 to 98.3), respectively, and low specificity of 17.1% (95% CI 13.2 to 21.4) and 13% (95% CI 9.6 to 17). Negative SIRS score had an intermediate sensitivity and specificity (Table 3).

The AUROC curve of a negative Systemic Inflammatory Response Syndrome (SIRS), quick Sequential Organ Failure Assessment (qSOFA) score, and quick Pediatric Logistic Organ Dysfunction-2 (qPELOD-2) score is presented in Figure 2.

The AUROC curve of a negative SIRS for predicting no antibiotic approach tends to be higher than a negative qPELOD-2 score (0.59 vs. 0.54; *p*-value = 0.023). However, the AUROC curve between SIRS and qSOFA (0.59 vs. 0.56; *p*-value = 0.262) as well as between qSOFA and qPELOD-2 (0.56 vs. 0.54; *p*-value = 0.071) were comparable (Table 3).

### 2.3. Factors Associated with No Antibiotic Approach in PICU

Using logistic regression models adjusted for potential confounders, the relative odds of no antibiotic approach increased in older children and the presence of hypernatremia. However, the relative odds decreased in the presence of severe underweight, altered mentation, age-specific fast breathing, lower chest wall indrawing, adventitious sound on lung auscultation, abdominal distension, developmental delay, hyponatremia, hypocalcemia, and microscopic evidence of invasive diarrhea (Table 4).

## 3. Discussion

In this retrospective case-control study of 510 children aged 2 to 59 months admitted to an ICU of an LMIC, we analyzed the accuracy of negative Systemic Inflammatory Response Syndrome (SIRS), quick Sequential Organ Failure Assessment (qSOFA) and quick Pediatric Logistic Organ Dysfunction-2 (qPELOD-2) scores for implementing a no antibiotic approach in PICU. We also focused on the associated factors and outcome of children who did not receive antibiotics.

Some noteworthy observations were: first, we observed both negative qSOFA and qPELOD-2 scores showed higher sensitivity to identify children suitable for implementing a no antibiotic approach, although the observations were compromised by lower specificity. Second, the no antibiotic approach was more often implemented in older children and children with hypernatremia. Third, this approach is implemented less often in children with severe underweight, altered mentation, age-specific fast breathing, chest wall in-drawing, adventitious sound on lung auscultation, abdominal distension, developmental delay, hyponatremia, hypocalcemia and microscopic evidence of invasive diarrhea.

Although clinicians often struggle to decide when to initiate antibiotics in critically ill children in ICU, still no study has examined the utility of negative SIRS, qSOFA, and qPELOD-2 scores as a bedside screening tool to overcome this challenging situation. We found negative qSOFA and qPELOD-2 with <2 scores have high sensitivity of 96% for predicting a no antibiotic approach, though no scoring system has high specificity.

However, considering AUROC, negative SIRS (<2 scores) is equivalent to qSOFA and more accurate than qPELOD-2. A previous study examining the performance of qSOFA demonstrated that the prevalence of PICU transfer and or mortality was 2% with qSOFA < 2 scores compared to 22.5% with qSOFA ≥ 2 in children presented to the emergency department [21]. The sepsis scoring systems have been developed to identify infection prompting immediate antibiotic therapy, though they lack sufficient sensitivity and specificity to capture sepsis. A prior study showed that the sensitivity and specificity of qSOFA were ≥37% and 79%, respectively, and that of SIRS were ≥80% and 21%, respectively, indicating neither scoring system truly identifies children requiring antibiotic therapy [22].

We found that antibiotic prescriptions were less in relatively older children. In this context, young infants are at increased risk for severe infection and often receive empirical antibiotic therapy. Rogawski et al. showed that antibiotic use was common in the first 6 months of life, even at the community level [23]. The association of hypernatremia with no antibiotic approach in young diarrheal children is understandable. On admission, the most common cause of hypernatremia is excess water loss due to acute watery diarrhea [24,25], and viral infections are responsible for most watery diarrhea cases. Empirical antibiotic treatment must be judiciously considered against inadvertent and potentially injurious consequences in such children [26,27].

Similarly, children showing microscopic evidence of invasive diarrhea and dysentery (visible blood in stool) should be treated with antibiotics [28] as *Shigella* infection, associated with considerable mortality and morbidity [29], is suspected. However, eight children had microscopic evidence of invasive diarrhea, and three had bacterial isolates from their stool sample among children in the no antibiotic group. These children had not been treated with antibiotics because their diarrhea resolved spontaneously before the availability of reports.

None of our cases had SAM. Several studies have revealed a high prevalence of infections among children hospitalized for SAM, and routine antibiotic therapy is recommended for such children [30]. Children with severe underweight (WAZ < −3 but >−4) seem to be at considerably higher risk of infection compared to well-nourished children [31,32]. A meta-analysis of more than five thousand children showed substantially higher mortality in underweight and wasted children than well-nourished children [33]. We also observed less frequent implication of a no antibiotic approach in children with severe underweight and stunting.

The association of less implication of a no antibiotic approach in children with age-specific fast breathing, chest wall in-drawing, adventitious sounds on lung auscultation is understandable. In LMIC, the diagnosis of pneumonia is based on cough and or breathing difficulty with clinical signs such as fast breathing and lower chest wall in-drawing. The presence of hypoxemia is a sign of severe pneumonia [34]. World Health Organization (WHO) recommended routine antibiotic therapy for childhood pneumonia to reduce mortality. In this study, the proportion of fast breathing and chest in-drawing among children who did not receive antibiotics was 15% and 2%, respectively, and none had a cough. Additionally, none of them had radiological evidence of pneumonia. These clinical signs without having a cough are known to be a nonspecific diagnostic tool for pneumonia, as most of our children had diarrhea, and our study children might demonstrate these signs mainly due to metabolic acidosis [35,36]. We found two children in the no antibiotic group had hypoxemia due to congenital cyanotic heart disease, so we did not include hypoxemia in the logistic regression model.

We found that children who did not receive antibiotics less often had altered mentation, abdominal distension, hypocalcemia, and hyponatremia. Earlier studies have shown that altered mentation and abdominal distension in critically sick children might represent sepsis [37] and should be treated with antibiotics. Studies have shown that total and ionized calcium significantly reduced in sepsis. The mechanism of sepsis-induced hypocalcemia remains unknown; however, this appears to be associated with elevated levels of proinflammatory cytokines, such as tumor necrosis factor (TNF)-*α*, interleukin (IL)-1, IL-6; hypoparathyroidism, vitamin D deficiency, or resistance [38]. Similarly, hyponatremia is commonly encountered in critically ill children with respiratory tract infections, sepsis and CNS infections. In a recent study, Park SW et al. reported that co-infection with multiple pathogens was more frequent in hyponatremic children than in those without hyponatremia [39].

### Limitation

Our study result might not be generalizable because it was conducted in a single-center PICU of a diarrheal disease hospital. We comprehend that this is an important limitation of the study since results may not be extrapolated to other PICUs where patients with other syndromes are admitted. In developing countries, the PICU mortality ranges from 13 to 25% [40,41], though we observed relatively fewer deaths (5%) in our cohort. We do not have a clear explanation for this, but it is likely that patients might not be as severe as in multi-disciplinary hospitals. However, in 2007, the mortality rate of our ICU was 11% [42]. The introduction of bubble CPAP therapy as a standard of care for severe pneumonia with hypoxemia in under-five children since 2013 may have contributed to better outcomes [43]. Additionally, scrupulous adherence to management guidelines of SAM and other treatment protocols may impact this low mortality. The retrospective nature of the study is another important limitation.

## 4. Materials and Methods

### 4.1. Study Site

We conducted this study in a single center diarrheal-hospital, the Dhaka Hospital of the International Centre for Diarrhoeal Disease Research, Bangladesh (icddr,b), located in Dhaka, Bangladesh. This hospital is the largest diarrheal disease hospital globally and provides care for approximately 150,000 patients annually. Among them, sixty-two percent are aged below five years. Patients presenting with diarrhea with or without other comorbidities can seek health care at this hospital. The emergency department has a well-organized triage for sorting and prioritizing patients for care and initiating appropriate therapeutic measures. More than 90% of patients are managed in the short-stay ward, where the median duration of stay is 18 h. Only 1–2% of individuals needing immediate resuscitation at triage or patients with a critical illness or those with clinical deterioration in the short-stay ward or longer-stay ward are transferred to the ICU. The median duration of stay at the ICU is five days.

### 4.2. Study Population and Design

We conducted a retrospective observational study with an unmatched case-control design. Children aged 2–59 months admitted to the ICU of Dhaka Hospital with suspected critical illness between January 2017 and January 2020 were eligible for enrolment. We retrieved data through the hospital’s electronic database of the patient information system. Patients were considered to have critical illness if they met ICU transfer criteria following hospital guidelines (Appendix A) and stayed in ICU for more than 24 h. The cases were children who did not receive an antibiotic during their ICU stay and were discharged without antibiotic therapy. Children who received oral or parenteral antibiotics at triage or following ward transfer before admission in the ICU or during discharge were excluded.

In contrast, controls were children who received one or more intravenous antibiotic therapy during their stay at ICU. We assumed 85% exposure of acute watery diarrhea (AWD) in controls to compare the percentage of cases and controls, considering the high prevalence of AWD among children with high infection burden countries [44]. To provide 80% power at a 5% level of significance (two-sided) and desired odds ratio (OR) of 2, we aimed to enroll 164 cases and 346 controls. Controls were selected using computer-generated random number sequences using SPSS version 20.0 for windows. The database identified 1821 controls, and 1:2 unmatched case-control ratios were used to increase the statistical power.

### 4.3. Measurement

We developed a pretested case report form for the acquisition of relevant data. We reviewed all medical records, including the initial presentation to the triage, inpatient course, and outcome. Data on demographics (age, gender, breastfeeding, immunization as per EPI schedule), vital signs (temperature, heart rate, respiratory rate, mean arterial pressure, and SpO2), anthropometric measurements, clinical features (duration and consistency of diarrhea, duration of vomiting, dehydration status, documented seizure, altered mentation, developmental delay, congenital heart disease) and outcome were collected. We also examined laboratory results, including complete blood count, hypernatremia, hyponatremia, hyperkalemia, hypokalemia, metabolic acidosis, hypocalcemia, microscopic evidence of invasive diarrhea, UTI, and CNS infection, as well as bacterial pathogens from blood and stool culture for those helped with a no antibiotic approach.

### 4.4. Scoring Systems

We calculated SIRS, qSOFA, and qPELOD-2 scores based on the first measured values after ICU entry. SIRS [15] was defined as tachycardia, age-specific fast breathing, temperature abnormality (axillary temperature > 38 °C or <36 °C ), and white blood cell abnormality (>12,000/mm^3^ or <4000/mm^3^ or bandemia >10%) in the absence of dehydration. SIRS score must include either temperature or white blood cell abnormality. qSOFA [45] criteria were: hypotension, age-specific fast breathing, and altered mentation. qPELOD-2 [19] criteria were tachycardia, hypotension, and altered mentation. A threshold of fewer than two scores was applied to indicate a negative result for every score. We defined altered mentation as having drowsiness, disorientation, confusion or coma. We defined age-specific fast breathing if the respiratory rate was >50 breaths/min for 2–11 months and >40 breaths/min for 12–59 months; tachycardia if the heart rate was ≥160 beats/min for 2–11 months, ≥140 beats/min for 12–59 months; and hypotension if mean arterial pressure <50 mm Hg.

### 4.5. Management

All children admitted to ICU were initially managed by trained ICU physicians following hospital guidelines. All children received standard treatment, including noninvasive or invasive oxygen therapy, intravenous fluid, suitable antibiotics (if necessary), anti-seizure therapy for seizure, and other supportive management. We adopted a no antibiotic approach among the children who did not have severe acute malnutrition, sepsis [46], severe pneumonia (following WHO classification). During the initial clinical assessment, we evaluated history, physical examination and simultaneously quick laboratory tests (complete blood count, stool for microscopic examination in diarrheal children, microscopic examination of urine and study of cerebrospinal fluid in children having a seizure) to identify any focus of bacterial infection. Attending physicians evaluated patients every 8 h, and ICU consultants made clinical rounds at least twice a day. Antibiotics were added if the patient’s condition deteriorated clinically, as discerned by the attending physician.

### 4.6. Definition

We defined severe stunting with a length or height for age z-score (LAZ/HAZ) ˂ −3, severe underweight with weight for age z-score (WAZ) < −3 but ≥−4 ensuing WHO growth standards [47]. We defined severe acute malnutrition (SAM) with a weight for length or height z score (WHZ) < −3 or weight for age z score (WAZ) <−4 of the median of the WHO anthropometry or presence of nutritional edema [48]. We defined hypernatremia and hyponatremia if serum sodium concentration was >150.0 and <130.0 mmol/L; hyperkalemia and hypokalemia if serum potassium concentration was >5.5 and <3.5 mmol/L; metabolic acidosis if serum TCO_2_ was <17.0 mmol/L, hypocalcemia if serum calcium was <2.12 mmol/L.

### 4.7. Statistical Analysis

We analyzed data using the Stata v13.0 for Windows (Stata Corp LLC, College Station, TX, USA). We assessed the normality of distribution with the Shapiro–Wilk test. Continuous variables were expressed as mean ± SD or median (IQR), categorical variables as frequency and percentages. When comparing the no antibiotic and antibiotic groups, Student’s t-test or Mann–Whitney’s *U* test, as appropriate, was used to analyze the continuous variables. The Chi-square test or Fisher’s exact test was used to compare differences in proportions. We applied multiple logistic regression analysis to identify the no antibiotic approach’s determinants and the evolution of variables having a *p*-value < 0.05 at the bivariate analysis; OR were adjusted, and 95% confidence intervals were calculated.

The receiver operating characteristics (ROC) curve was created for each score. We determined each scoring system’s predictive accuracy using the AUROC (area under the receiver operating characteristics curve). We calculated sensitivity, specificity, positive and negative predictive values using contingency tables for every score. We compared the sensitivity and specificity of negative SIRS, qSOFA, and qPELOD-2 scores for no antibiotic approach using McNemar’s test and the AUROC curve using DeLong’s method [49]. A *p*-value of <0.05 was considered statistically significant.

## 5. Conclusions

This is the first study in the epoch of antibiotic resistance to determine the utility of bedside screening tools and associated factors that might help PICU clinicians avoid overusing antibiotics in children with diarrhea and other comorbidities, especially in resource-poor settings. Based on our findings, we may propose that PICU clinicians adopt the no antibiotic or watchful waiting approach in children with a negative (score < 2) qSOFA or qPELOD-2 score calculated during PICU admission. However, in addition to these simple screening tools, physicians should judiciously implement a no antibiotic approach by careful diagnosis and severity assessment based on underlying pathophysiology to limit harmful consequences in PICU. However, as this result may be applicable for children with diarrhea in PICU, future prospective studies involving larger populations in different settings may consolidate our observation.

## Figures and Tables

**Figure 1 antibiotics-10-01255-f001:**
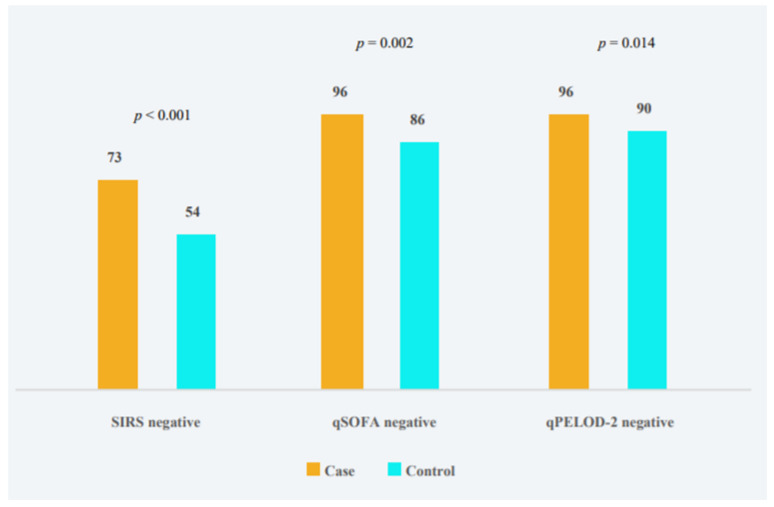
Distribution of included children without and with antibiotic therapy according to the percentage of negative (<2 scores) SIRS, qSOFA and qPELOD-2 criteria met.

**Figure 2 antibiotics-10-01255-f002:**
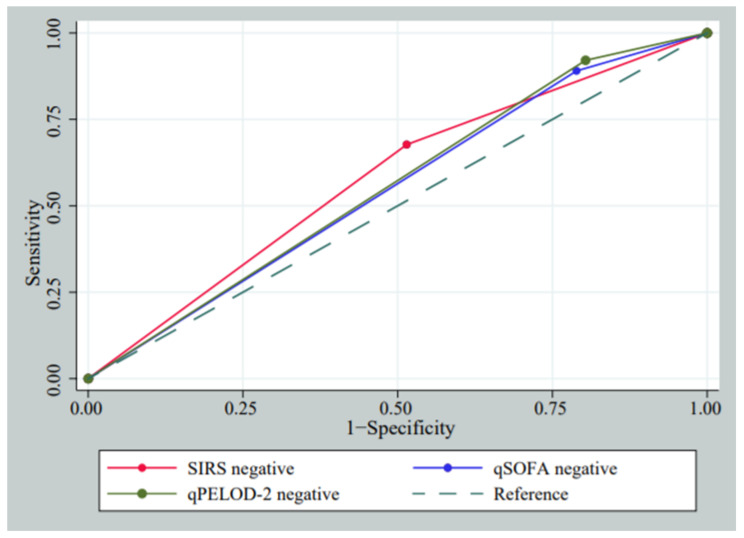
ROC curves of negative (<2 scores) SIRS, qSOFA, and qPELOD-2 criteria for predicting no antibiotic therapy at ICU. ICU, Intensive Care Unit; SIRS, Systemic Inflammatory Response Syndrome; qSOFA, quick Sequential Organ Failure Assessment; qPELOD-2, quick Pediatric Logistic Organ Dysfunction score-2; ROC, Receiver Operating Characteristic curve.

**Table 1 antibiotics-10-01255-t001:** Sociodemographic and clinical characteristics of critically ill ICU children aged below five years without (cases) and with antibiotic therapy (controls) in Bangladesh.

Characteristics	Children without Antibiotic (n = 164)	Children with Antibiotic (n = 346)	OR	95% CI	*p*-Value
Male	104 (63)	211 (61)	1.11	(0.75–1.63)	0.598
Age in months (median, IQR)	10.1 (7.1, 14.0)	8.1 (5.4, 12.3)	-		0.002
Breast feed	108 (65)	218 (63)	1.13	(0.77, 1.67)	0.532
Immunized as per EPI schedule	127 (77)	261 (75)	1.12	(0.72, 1.74)	0.620
SAM	0	119 (23)	-	-	-
Severe stunting	7 (4)	86 (25)	0.13	(0.06, 0.30)	<0.001
Severe under weight	11 (7) *	132 (38)	0.12	(0.06, 0.22)	<0.001
Watery stool	148 (90)	287 (83)	1.90	(1.06, 3.42)	0.030
Duration of diarrhea	152/164, 3.0 (2.0, 5.0)	313/346, 3.0 (2.0, 5.0)			0.193
History of vomiting	44 (27)	66 (19)	1.56	(1.00, 2.41)	0.048
Dehydration	39 (24)	123 (36)	0.56	(0.37, 0.86)	0.008
Fever	36 (22)	97 (28)	0.72	(0.47, 1.12)	0.149
Age specific fast breathing	53/157 (34)	176/340 (52)	0.47	(0.32, 0.70)	<0.001
Chest in-drawing	4 (2)	89 (26)	0.07	(0.03, 0.20)	<0.001
Adventitious sound on lung auscultation	1 (0.6)	119 (34)	0.01	(0.00, 0.08)	
Hypoxemia	2 (1) **	59 (17)	0.06	(0.01, 0.25)	<0.001
Abdominal distension	4 (2)	42 (12)	0.18	(0.06, 0.51)	<0.001
Seizure	84 (51)	106 (31)	2.37	(1.62, 3.48)	<0.001
Altered mentation	41 (25)	145 (42)	0.46	(0.30, 0.70)	<0.001
Development delay	5 (3)	38 (11)	0.25	(0.09, 0.66)	0.003
Congenital heart disease	3 (2)	23 (7)	0.26	(0.08, 0.88)	0.021
Death	0 (0)	8 (5.3)			0.044

* Weight for age z-score < −3 but ≥−4. ** Hypoxemia due to congenital cyanotic heart disease; SAM, severe acute malnutrition.

**Table 2 antibiotics-10-01255-t002:** Laboratory characteristics of critically ill ICU children aged below five years without (cases) and with antibiotic therapy (controls) in Bangladesh.

Characteristics	Case (n = 164)	Control (n = 346)	OR	95% CI	*p*-Value
Hypernatremia	53 (32)	66 (19)	2.03	(1.33, 3.09)	0.001
Hyponatremia	14 (9)	76 (22)	0.33	(0.18, 0.61)	<0.001
Hypokalemia	46 (28)	128 (37)	0.66	(0.44, 0.99)	0.047
Hyperkalemia	7 (4)	44 (13)	0.31	(0.13, 0.69)	0.003
Acidosis	92 (56)	210 (61)	0.83	(0.57, 1.21)	0.324
Hypocalcemia	36 (22)	115 (33)	0.56	(0.37, 0.87)	0.009
Anemia	87 (53)	184 (53)	0.99	(0.68, 1.44)	0.978
Leukocytosis	75/140 (54)	203/318 (64)	0.65	(0.44, 0.98)	0.038
Microscopic evidence of invasive diarrhea	8 (5)	45 (13)	0.34	(0.16, 0.75)	0.005
Laboratory evidence of CNS infection	0	9 (3)	-		0.029
Bacteremia	0	41 (12)	-	-	<0.001
Bacterial isolates from stool culture	3/17 (18)	41/177 (23)	0.71	(0.19, 2.59)	0.767

**Table 3 antibiotics-10-01255-t003:** Prognostic accuracy of scoring systems (95%CI), area under the curve (95% CI) and comparison to AUROC of scoring systems (95%CI) for predicting no antibiotic approach among under-five children in ICU, Bangladesh.

* Scores	Sensitivity% (95% CI)	Specificity% (95% CI)	NPV% (95% CI)	PPV% (95% CI)	AUROC (95% CI)	Comparison to AUROC	*p*-Value
Negative SIRS	72.1 −(64.8, 78.7)	46.0 −(40.6, 51.4)	76.8 −(70.5, 82.4)	39.9 −(34.4, 45.5)	0.59 −(0.55, 0.63)	Negative SIRS vs. qPELOD-2	0.023
Negative qSOFA	95.9 −(91.8, 98.3)	17.1 −(13.2, 21.4)	89.4 −(79.4, 95.6)	36.5 −(32.1, 41.1)	0.56 −(0.54, 0.59)	Negative SIRS vs. qSOFA	0.262
Negative qPELOD-2	95.9 −(91.8, 98.3)	13.0 −(9.6, 17.0)	86.5 −(74.2, 94.4)	35.4 −(31.1, 39.9)	0.54 −(0.52, 0.57)	Negative qSOFA vs. qPELOD-2	0.071

* All analyses used thresholds of SIRS < 2, qSOFA < 2, and qPELOD-2 score < 2. AUROC, area under the receiver operating characteristics curve; ICU, intensive care unit; SIRS, Systemic Inflammatory Response Syndrome; qSOFA, quick Sequential Organ Failure Assessment; qPELOD-2, quick Pediatric Logistic Organ Dysfunction Score-2; NPV, negative predictive value; PPV, positive predictive value.

**Table 4 antibiotics-10-01255-t004:** Logistic regression analysis revealing the independently associated factors for adopting a no antibiotic approach in under-five ICU children in Bangladesh.

Characteristics	OR	(95% CI)	*p*-Value
Male sex	1.44	(0.82, 2.54)	0.202
Age in months	1.03	(1.01, 1.07)	0.024
Hypernatremia	2.64	(1.31, 5.31)	0.006
Hyponatremia	0.39	(0.17, 0.86)	0.021
Hypocalcemia	0.46	(0.24, 0.89)	0.018
Hyperkalemia	0.44	(0.15, 1.25)	0.122
Acute watery diarrhea	1.08	(0.45, 2.61)	0.856
Dehydration	0.75	(0.39, 1.46)	0.401
Severe stunting	0.62	(0.19, 1.99)	0.423
Severe underweight	0.23	(0.08, 0.65)	0.005
Altered mentation	0.43	(0.23, 0.80)	0.008
History of vomiting	1.86	(0.96, 3.61)	0.066
Age specific fast breathing	0.88	(0.50, 1.55)	0.653
Lower chest wall in-drawing	0.16	(0.04, 0.61)	0.008
Adventitious sound on lung auscultation	0.22	(0.00, 0.17)	<0.001
Abdominal distension	0.18	(0.05, 0.60)	0.005
Documented seizure	1.50	(0.85, 2.65)	0.156
Development delay	0.32	(0.09, 1.10)	0.072
Congenital heart disease	0.39	(0.03, 4.65)	0.462
Microscopic evidence of invasive diarrhea	0.19	(0.07, 0.48)	<0.001
Leukocytosis	0.94	(0.53, 1.67)	0.841

## Data Availability

Data are available from the corresponding authors on reasonable request.

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
