# Peer review of "The Utility of Bedside Assessment Tools and Associated Factors to Avoid Antibiotic Overuse in an Urban PICU of a Diarrheal Disease Hospital in Bangladesh"

_antibiotics, 2021, doi:10.3390/antibiotics10101255_

Round 1

Reviewer 1 Report

The topic of this original article is interesting, and the manuscript is well written. However, due to the characteristics of the study, the results cannot be extrapolated to other centers. The external validity and generalizability of the results are a moot point. Major problems of the study:

1) The study is conducted in a monographic hospital where patients presenting diarrhea are admitted.  I think it is an important limitation of the study since results may not be extrapolated to other PICUs where patients with other syndromes are admitted.  In fact, one of the most frequent etiologies of diarrhea in LMICs countries are viral diarrhea; therefore, the article conclusions may not be applicable to other syndromes.  The characteristics of the center (single diarrheal-hospital center) should be highlighted more in the article so that the reader can take it into account.

2) Case group (no antibiotic group) and control group (antibiotic) are unbalanced and not comparable in the main outcome. As authors suggest, water diarrhea is more frequent in case group and its etiology is viral. In contrast, in control group is significantly more frequent leukocytosis, severe illness, microscopic evidence of invasive diarrhea and isolation of bacterial pathogens. As study is evaluating the “no antibiotic approach” in PICU, conclusions may be dangerous if extrapolated to another type of patient or other kind of pathology.

3) The study has two main objectives: 1) to evaluate the accuracy of negative qSOFA, and qPELOD-S scores in predicting no antibiotic approach and 1) to identify the associated factors and outcomes of implementing a no antibiotic approach at PICU. In my opinion, due to the characteristics of the study and the differences between case and control groups the second objective cannot be evaluated correctly.

4)  Moreover, one of the conclusions of the study is that case group had not more mortality than control group (liens 336-338). However, patients in control group are significantly more severe patients (younger patients, more severe underweight, more altered mentation, higher respiratory rate…)

Minor issues:

1) Abstract is not clear and confusing, it should be restructured and rewritten

2) Please clarified in abstract what cases means (not receiving antibiotic) and what control means (receiving antibiotics).

3) Introduction is well-written but sometimes redundant and it can be summarized.

4) Please, consider eliminating lines 46-49 that are not relevant and quite confusing.

5) SAM should be defined in the main text of the manuscript.

Author Response

Response to reviewer: 1

The topic of this original article is interesting, and the manuscript is well written. However, due to the characteristics of the study, the results cannot be extrapolated to other centers. The external validity and generalizability of the results are a moot point. Major problems of the study:

Response: We greatly appreciate the valuable and precise observations from the respected reviewer and have tried to address them. However, we agree that due to the study's characteristics, the results cannot be extrapolated to other centers. Diarrhea is the second leading cause of mortality among under-5 children globally. This is the first study examining the utility of simple bedside assessment tools and associated factors to avoid antibiotic overuse in PICU of a developing country, particularly in children with diarrhea. Thus the result of this study would be informative for PICU physicians of LMIC to limit antibiotic overuse and thus prevent antimicrobial resistance.

1) Comment: The study is conducted in a monographic hospital where patients presenting diarrhea are admitted.  I think it is an important limitation of the study since results may not be extrapolated to other PICUs where patients with other syndromes are admitted.  In fact, one of the most frequent etiologies of diarrhea in LMICs countries are viral diarrhea; therefore, the article conclusions may not be applicable to other syndromes.  The characteristics of the center (single diarrheal-hospital center) should be highlighted more in the article so that the reader can take it into account.

Response: We appreciate our respected reviewers comment. This is a very important observation. We agree that we conducted this study in a diarrheal disease hospital where all children admitted with diarrhea with or without other complications. Thus the results may be applicable for children with diarrhea only. The result may not be applicable for other population. We mentioned this limitation in the Limitation section in line 277-279. We highlighted the characteristics of the center (single diarrheal-hospital center) in the article line 291-293. The characteristics of the center also highlighted in the title so that the reader can take it into account.

Title: ‘The utility of bedside assessment tools and associated factors to avoid antibiotic overuse in an urban PICU of a diarrheal disease hospital in Bangladesh

‘Our study result might not be generalizable because it was conducted in a single-center ICU of a diarrheal hospital. We comprehend that this is an important limitation of the study since results may not be extrapolated to other PICUs where patients with other syndromes are admitted.’

‘We conducted this study in a single center diarrheal-hospital, the Dhaka Hospital of the International Centre for Diarrhoeal Disease Research, Bangladesh (icddr,b), located in Dhaka, Bangladesh. This hospital is the largest diarrheal disease hospital globally and provides care for approximately 150,000 patients annually’.

2) Comment: Case group (no antibiotic group) and control group (antibiotic) are unbalanced and not comparable in the main outcome. As authors suggest, water diarrhea is more frequent in case group and its etiology is viral. In contrast, in control group is significantly more frequent leukocytosis, severe illness, microscopic evidence of invasive diarrhea and isolation of bacterial pathogens. As study is evaluating the “no antibiotic approach” in PICU, conclusions may be dangerous if extrapolated to another type of patient or other kind of pathology.

Response: Thanks for this valuable opinion. We revised the conclusion lines 392-395 and 398-404, which are applicable only for children with diarrhea. We also revised the conclusion in the abstract. However, both the groups had diarrhea and other comorbidities and ended up with PICU admission. Although we found watery diarrhea was more frequent among cases in the bivariate analysis, the logistic regression analysis did not show an independent association of watery diarrhea with no antibiotic group. However, we quite agree with the respected reviewer that the conclusion should be more focused on what we have done now.

‘This is the first study in the epoch of antibiotic resistance to determine the utility of bedside screening tools and associated factors that might help PICU clinicians avoid overusing antibiotics in children with diarrhea and other comorbidities, especially in resource-poor settings. Based on our findings, we may propose that PICU clinicians adopt no antibiotic or watchful waiting approach in children with a negative (score <2) qSOFA or qPELOD-2 score calculated during PICU admission. However, in addition to these simple screening tools, physicians should judiciously implement a no antibiotic approach based on the clinical diagnosis and underlying pathophysiology to limit harmful consequences in PICU. However, as this result may be applicable for children with diarrhea in PICU, future prospective studies involving larger populations in different settings may consolidate our observation.’

3) Comment: The study has two main objectives: 1) to evaluate the accuracy of negative qSOFA, and qPELOD-S scores in predicting no antibiotic approach and 1) to identify the associated factors and outcomes of implementing a no antibiotic approach at PICU. In my opinion, due to the characteristics of the study and the differences between case and control groups the second objective cannot be evaluated correctly.

Response: We appreciate the valuable opinion of our respected reviewer. We revised our objectives based on our respected reviewers’ suggestions in lines 90-92. Thus, revised the result (line 137-141; 167 173) and discussion section (line 178-209) accordingly. We can understand the concern raised by our respected reviewer. As both case and control groups had diarrhea and were admitted to the PICU with several comorbidities; thus the characteristics, associated factors, and outcomes of implementing a no antibiotic approach at PICU in a LMIC need to be emphasized in this patient population to prevent antimicrobial resistance (AMR), one of the most important public health threats worldwide. Thus, we did not drop the second objective.

4)  Comment: Moreover, one of the conclusions of the study is that case group had not more mortality than control group (liens 336-338). However, patients in control group are significantly more severe patients (younger patients, more severe underweight, more altered mentation, higher respiratory rate…)

Response: We appreciate our respected reviewer's comment. We dropped this line from the conclusion and revised accordingly (399-401). However, we included children who required PICU admission in a developing country. Among them, we identified children who were treated with antibiotics and who did not require antibiotics. Based on that, we aimed to identify associated factors of children who did not require antibiotics. Our results reflect that our cases did not have sepsis or pneumonia, or other bacterial infections and we optimize treatment by careful diagnosis and severity assessment. This confirms the justification of not giving antibiotics to children who presented with diarrhea and other complications to the PICU. This study would be informative for PICU physicians of LMIC where injudicious use of antibiotics is a common practice, as discussed in the introduction section.

Minor issues:

1) Comment: Abstract is not clear and confusing, it should be restructured and rewritten

Response: Thank you for this valuable observation. We restructured the abstract and written accordingly

2) Comment: Please clarified in abstract what cases means (not receiving antibiotic) and what control means (receiving antibiotics)

Response: We appreciate our respected reviewer’s suggestion and defined cases and controls in the abstract

3) Comment: Introduction is well-written but sometimes redundant and it can be summarized.

Response: We appreciate our respected reviewers comment. We eliminated lines 63-65 and revised line 79.

4) Comment: Please, consider eliminating lines 46-49 that are not relevant and quite confusing.

Response: Thank you. We eliminated lines 46-49 from introduction.

5) Comment: SAM should be defined in the main text of the manuscript.

Response: Thank you. We defined SAM in the 4.6. Definition section (line 367-368).

Reviewer 2 Report

Dear Authors,

This is a very interesting and important study. It is very well written and well presented. Actually, I only have some minor remarks, such as:

In the Introduction (l. 46-48) please specify which country was this study conducted in.

In Material and Methods

l. 279: mm3 should be in fact mm3?

Also, in my opinion, there are too many abbreviations used in the text, which makes it sometimes difficult to follow the merit. I wonder whether it is possible to use the full terms in some cases or limit the use of abbreviations (particularly in the Results section).

Author Response

This is a very interesting and important study. It is very well written and well presented. Actually, I only have some minor remarks, such as:

Response: We greatly appreciate the valuable and precise observations from the respected reviewer and have tried to address them.

Comment: In the Introduction (l. 46-48) please specify which country was this study conducted in.

Response: Thank you, we reviewed the article thoroughly but did not find the name of the country, the study was conducted in PICU of a Tertiary Hospital. The article was published in Indian Journal of Critical Care Medicine. However, as one of our respected reviewer suggested to eliminating lines 46-49, we eliminated these lines and the reference (8) from the introduction.

Comment: In Material and Methods: l. 279: mm3 should be in fact mm3?

Response: We apologize for this typographical error and corrected accordingly in line 340.

‘(>12000/mm3 or <4000/mm3 or bandemia >10%)’

Comment: Also, in my opinion, there are too many abbreviations used in the text, which makes it sometimes difficult to follow the merit. I wonder whether it is possible to use the full terms in some cases or limit the use of abbreviations (particularly in the Results section).

Response: We appreciate our respected reviewers comment. We tried to limit the use of abbreviations particularly in the Results section. We also limited the use of abbreviations in discussion section.

Result: ‘Of the included children, the frequency of Systemic Inflammatory Response Syndrome (SIRS), quick Sequential Organ Failure Assessment (qSOFA) score and quick Pediatric Logistic Organ Dysfunction Score-2 (qPELOD-2) negativity was 311/510 (61%), 452/510 (89%), and 475/510 (91%), respectively’.

Discussion: ‘In this retrospective case-control study of 510 children aged 2 to 59 months admitted to an ICU of an LMIC, we analyzed the accuracy of negative Systemic Inflammatory Response Syndrome (SIRS), quick Sequential Organ Failure Assessment (qSOFA) and quick Pediatric Logistic Organ Dysfunction-2 (qPELOD-2) scores for implementing a no antibiotic approach in PICU’.

Round 2

Reviewer 1 Report

Accept in the present form.